# Nisin Inhibition of Gram-Negative Bacteria

**DOI:** 10.3390/microorganisms12061230

**Published:** 2024-06-19

**Authors:** Adam M. Charest, Ethan Reed, Samantha Bozorgzadeh, Lorenzo Hernandez, Natalie V. Getsey, Liam Smith, Anastasia Galperina, Hadley E. Beauregard, Hailey A. Charest, Mathew Mitchell, Margaret A. Riley

**Affiliations:** 1Department of Biology, University of Massachusetts, Amherst, MA 01002, USA; acharest@umass.edu (A.M.C.); ereed@umass.edu (E.R.); sbozorgzadeh@umass.edu (S.B.); lrhernandez@umass.edu (L.H.); ngetsey@umass.edu (N.V.G.); liamsmith@umass.edu (L.S.); agalperina@umass.edu (A.G.); hbeauregard@umass.edu (H.E.B.); hcharest@umass.edu (H.A.C.); 2Organicin Scientific, 240 Thatcher Road, Amherst, MA 01003, USA; mjm635@georgetown.edu

**Keywords:** bacteriocins, bacteria, nisin, antimicrobials, alternatives to antibiotics, antimicrobial resistance

## Abstract

**Aims:** This study investigates the activity of the broad-spectrum bacteriocin nisin against a large panel of Gram-negative bacterial isolates, including relevant plant, animal, and human pathogens. The aim is to generate supportive evidence towards the use/inclusion of bacteriocin-based therapeutics and open avenues for their continued development. **Methods and Results:** Nisin inhibitory activity was screened against a panel of 575 strains of Gram-negative bacteria, encompassing 17 genera. Nisin inhibition was observed in 309 out of 575 strains, challenging the prevailing belief that nisin lacks effectiveness against Gram-negative bacteria. The genera *Acinetobacter*, *Helicobacter*, *Erwinia*, and *Xanthomonas* exhibited particularly high nisin sensitivity. **Conclusions:** The findings of this study highlight the promising potential of nisin as a therapeutic agent for several key Gram-negative plant, animal, and human pathogens. These results challenge the prevailing notion that nisin is less effective or ineffective against Gram-negative pathogens when compared to Gram-positive pathogens and support future pursuits of nisin as a complementary therapy to existing antibiotics. **Significance and Impact of Study:** This research supports further exploration of nisin as a promising therapeutic agent for numerous human, animal, and plant health applications, offering a complementary tool for infection control in the face of multidrug-resistant bacteria.

## 1. Introduction 

A staggering 19 million kilograms of antibiotics are used in the U.S each year, of which 65% or more are devoted to agriculture and animal health [1]. One outcome of such use is the emergence and rapid distribution among bacterial species of hundreds of unique genes conferring antibiotic resistance, thus contributing to the already challenging presence of a robust antibiotic resistome [2]. Human pathogens have easy access to this resistance reservoir, which is found in our own microbiomes and in our food sources, and is also widely distributed in the environment [3]. To slow the growth of the resistome and the corresponding levels of multi-drug resistant pathogens that acquire these genes, we must reduce our reliance on conventional antibiotics and seek both alternative and complementary approaches to infection prevention and treatment. 

One highly promising class of potential drug candidates is bacteriocins, which are peptides or protein antimicrobials produced by and active against most if not all, species of bacteria [4]. Bacteriocins are known for their relatively narrow killing spectra, typically targeting close relatives of the producing species. However, more broad antibacterial activity has been seen for numerous bacteriocins, including garvicin KS, nisin, and bactofencin A [5,6]. The functional diversity of bacteriocins [7] and their relative lack of toxicity against animal and plant cells have been understood for some time [8]. Further, there is a growing body of literature where bacteriocins are being assessed for potential efficacy in human health implications. One such study involved treating a small sample of women suffering from *Staphylococcus*-based mastitis with nisin, which revealed that nisin effectively cleared an infection that the traditional antibiotic treatments of choice, including cloxacillin, clindamycin, and/or erythromycin, could not [9]. Additionally, nisin has been shown to be an effective novel antimicrobial in the clinical treatment and prevention of *Streptococcus suis* infections in animals, as demonstrated in mouse trials [10]. In another in vivo study on mice, two nisin variants, nisin A and nisin V, were used to treat mice infected with *Listeria monocytogenes* and showed rapid infection control [11]. Similarly, increasing numbers of studies are focused on the development of bacteriocins for use in agriculture, such as the introduction of the gene clusters encoding the bacteriocins plantaricin and leucocin into tomato plants, which then confer resistance to several tomato plant pathogens [12]. 

One of the more widely known bacteriocins is nisin, a 34 amino acid peptide produced by *Lactococcus lactis* [13]. Nisin has been used in the food industry as a natural preservative for decades. It possesses high levels of activity against numerous pathogenic bacteria and low levels of toxicity for humans [14]. In fact, it was labeled GRAS (generally regarded as safe) for human consumption by the FDA in 1988 [15]. This designation led to its widespread use in food preservation with compelling activity against numerous foodborne pathogens, such as *Listeria monocytogenes*, *Clostridium perfringens*, *Clostridium botulinum*, and *Bacillus cereus* [16]. In addition, nisin is active against some of the more challenging Gram-positive human pathogens, including methicillin-resistant *Staphylococcus aureus*, multidrug-resistant *Streptococcus epidermidis,* and *Enterococcus* spp., with MICs at nanomolar concentrations [13,17]. 

Nisin targets the lipid II component of the bacterial cell wall (Figure 1A) [18,19]. It forms a complex with the lipid-II precursor that creates a thick peptidoglycan layer surrounding the cell and thus influences cell wall synthesis. Nisin also targets lipid II as a docking station for the formation of nm-sized pores, requiring four nisin–lipid II constituents and four additional nisin molecules. This binding process results in the creation of a pyrophosphate cage mediated by hydrogen bonds between nisin (eight molecules) and lipid II (four molecules), and results in the efflux of cellular constituents [20]. The interaction between nisin and lipid II is partly responsible for the low levels of nisin resistance observed in Gram-positive bacteria since lipid II is an essential component of the cell wall [20]. However, nisin resistance does occur through several mechanisms, including the production of peptidases that degrade it and modifications made at the membrane binding site [21].

When compared to the mass literature showing nisin activity against Gram-positive bacteria, there are relatively few such studies with Gram-negative bacteria, and the strain numbers employed are often quite limited [22]. This lack of attention is partially because the lipid II target of nisin in Gram-negative bacteria is sheltered by a relatively impermeable outer membrane, which excludes hydrophobic substances and macromolecules (Figure 1B). In fact, hydrophobic nisin has been shown to bind to the usually anionic surface of the Gram-negative outer membrane and stabilize the structure via electrostatic interactions [23]. In support of this proposed mechanical exclusion, the literature is replete with suggestions that nisin has limited activity against Gram-negative bacteria [17,23,24]. In those infrequent cases in which nisin sensitivity is reported [23], an alternative mode of action is proposed, one where nisin appears to crowd the outer membrane (OM) resulting in non-specific interactions with membrane lipid molecules (Figure 1B; steps a–b) [21]. This mode of action requires nisin to compete with cations to bind with the LPS, and it is interesting to note that many of the earlier studies of nisin activity employ growth media that provide high concentrations of cations, which may have been a confounding variable in determining its efficacy against Gram-negative genera [25,26]. Once the crowding of nisin creates a pore, the lipid II moieties become accessible and nisin interacts with them in a similar manner, as seen in the cell walls of Gram-positive bacteria (Figure 1B; step e). 

The compelling necessity for novel antimicrobials active against Gram-negative bacteria motivated us to reassess the potential for nisin to address this need. By 2050 it is predicted that antimicrobial resistance to currently used drugs could result in 10 million deaths per year taking the top spot over cancer as the leading cause of death [27]. The microbes reported to pose the highest threat to humankind as of 2019, according to the CDC, included numerous Gram-negative bacteria, including *Acinetobacter baumannii*, *Pseudomonas aeruginosa,* and *Salmonella* [28]. Here we present the results from a screen of nisin activity against a large, phylogenetically broad sample of Gram-negative bacteria, with the inclusion of multiple strains per genus. We employed a cation-rich nutrient medium (Luria–Bertani broth [29]) that was more conducive to the proposed mode of action of nisin against Gram-negative bacteria. The resulting data provides a more detailed inventory of nisin activity for a relatively understudied group of bacteria. 

## 2. Materials and Methods

### 2.1. Reagents 

Nisin A (93% pure) was obtained from ImmuCell (ImmuCell, Portland, ME, USA) and was stored at 4 °C. Luria–Bertani broth and agar were purchased from Fisher Bioreagents. For MIC assays, 11 two-fold serial dilutions of nisin were prepared in a citric acid buffer (0.01 M, pH 3.4) (Fisher Scientific, Fair Lawn, NJ, USA). The concentrations of nisin tested in this study ranged from 423 µg/mL to 0.2065 µg/mL.

### 2.2. Bacterial Strain Collection and Growth Standards 

569 strains were obtained from Professor Margaret Riley’s Environmental, Clinical, and Agricultural Strain Collection (University of Massachusetts Amherst) and the six *H. pylori* strains were generously donated from Dr. Scott Merrell’s collection (Uniformed Services University). Strains were grown in Luria–Bertani Broth (LB) at either 30 °C or 37 °C as appropriate and *H. pylori* strains were grown on heart brain infusion agar (Oxoid, Basingstoke, United Kingdom) with a fetal bovine serum additive (Sigma-Aldrich, St. Louis, MO, USA) and incubated in a Thermo Scientific™ AnaeroPack™-Anaero Anaerobic Gas Generator (Mitsubishi, Tokyo, Japan) at 37 °C. 

### 2.3. MIC of Gram-Negative Bacteria 

The minimum inhibitory concentration (MIC) of nisin for each strain was determined using a slight modification of the conventional broth-dilution method recommended by the Clinical and Laboratory Standards Institute [20]. In brief, bacteria were grown overnight in LB and diluted to a final concentration of approximately 1 × 10^5^ CFU/mL. Fifty μL of each strain was transferred to wells of 96 well microplates containing 130 μL of LB broth and 20 μL of each nisin concentration. The microplates were incubated at 37 °C for 24 h, and bacterial growth was assessed. The MIC values were defined as the lowest concentration of nisin that inhibited the visible growth of bacteria. Each assay was performed in triplicate. Rather than using the conventional MIC_90_ or MIC_50_ values to summarize and compare nisin MICs, a mean MIC was employed, due to the relatively limited sample sizes within some of the genera or species examined.

### 2.4. Data Quantification

The mean MIC was used to identify genera with high nisin inhibitory activity (mean MIC ≤ 26.4 μg/mL), moderate inhibitory activity (>26.4 μg/mL to <211.5 μg/mL), or low inhibitory activity (≥211.5 μg/mL). The percentage of sensitive strains was used to identify genera with broad nisin sensitivity (≥75% of strains sensitive), moderate sensitivity (≥25 to <75%), or narrow sensitivity (>1 to <25%). Clinical breakpoints have not been established for nisin, so we employed these arbitrary categories to group the levels of activity and sensitivity for ease of discussion.

### 2.5. Figure and Table Creation

For the creation of Table 1, Microsoft Word (Microsoft, Redmond, WA, USA) was used, for Figure 1 BioRender (BioRender, Toronto, ON, Canada) was used, for Figure 2 Microsoft Excel (Microsoft, Redmond, WA, USA) was used, and for Figure 3 R Studio (Version: 2023.12.1+402) using R release 3.3.0+ was used along with data package ggplot2.

## 3. Results 

### 3.1. Inhibitory Activity of Nisin against Gram-Negative Bacteria 

Table 1 provides a summary of nisin activity against a panel of 17 genera of Gram-negative bacteria. Of the 575 strains tested, 309 were found to have MICs within the range of nisin concentrations tested (423 to 0.2065 μg/mL).

For ease of comparison, Figure 2 provides a ranked order of sensitivity for the 17 genera examined. Four genera displayed sensitivity to nisin in all strains tested (*Acinetobacter*, n = 67; *Erwinia*, n = 21; *Helicobacter*, n = 5; *Xanthomonas*, n = 11). Only two possessed no detectable nisin activity (*Providencia*, n = 13 and *Morganella*, n = 13) at the highest concentration employed. Figure 3 is a violin plot of nisin MIC which provides a snapshot of the relative MIC values for each genus, and plots the median and quartile MIC values, as well as the distribution of strains along these values.

### 3.2. Genera with High Levels of Nisin Sensitivity 

Nearly 50% of genera tested showed what we define as high levels of nisin sensitivity, with ≥75% of the strains sensitive within the range of concentrations tested. All six *Helicobacter* strains tested were *H. pylori*, and were sensitive and had the lowest mean MIC detected (5.12 μg/mL). Similarly, 100% of the 67 strains of the *Acinetobacter* genera tested were *A. baumanii*, and they were sensitive, with a marginally higher mean MIC (5.86 μg/mL). One hundred percent of the fourteen strains of *Pseudomonas syringae* were sensitive, albeit with a significantly higher mean MIC of 27.18 μg/mL. The 21 strains of *Erwinia amylovora* were all sensitive, however, they required nearly twice as much nisin (mean MIC of 55.66 μg/mL). The 11 strains of *Xanthomonas*, which included five species (*X. maltophilia*, *X. campestris*, *X. axonopodis*, *X. vasicola*, and *X. citri)*, were all sensitive but required nearly three times more nisin (mean MIC of 130.36 μg/mL). Fifty of the fifty-one strains of *E. coli* tested show sensitivity, with a similarly high nisin mean MIC of 171.81 μg/mL. Thirty-nine of the forty-nine strains of *Citrobacter* (79.6%), which included four species (*C. koseri*, *C. amalonaticus*, *C. freundii*, *C. diversus*) were sensitive, with a mean MIC of 267.85 μg/mL. Nineteen of the twenty-four strains of *Vibrio* (79.2%), which included four species (*V. parahaemolyticus*, *V. alginolyticus*, *V. vulnificus*, and *V. cholerae)* were all sensitive, with a mean MIC of 60.13 μg/mL. 

### 3.3. Genera with Moderate Levels of Nisin Sensitivity 

Moderate levels of sensitivity were defined as ≥25% to <75% of the strains sensitive, which two genera exhibited. Twenty-four of the forty-one strains of *Serratia* (58.5%), representing three species (*S. plymuthica*, *S. marcescens*, and *S. rubidaea)*, were sensitive to nisin with a mean MIC of 371.0 μg/mL. Eleven of twenty-eight strains of *P. aeruginosa* (40%) were sensitive, with a mean MIC of 350 μg/mL. 

### 3.4. Genera with Low Levels of Nisin Sensitivity 

Low levels of sensitivity were defined as 1% to < 25% of the strains sensitive. Four genera fell into this category. Ten of the forty-eight strains of *Enterobacter* (20.8%) were sensitive, representing four species (*E. cloacae*, *E. taylorae*, *E. aerogenes*, and *E. agglomerans*) with a mean MIC of 296.76 μg/mL. Eight of the thirty-five strains of *Hafnia alvei* (20.5%) were sensitive with a mean MIC of 347.46 μg/mL. Nine of the sixty-five strains of *Klebsiella* (13.8%), which included two species (*K. pneumoniae* and *K. oxytoca)* were sensitive, with a mean MIC of 304.03 μg/mL. Two of the nineteen strains of *Burkholderia* (10.9%), which included four species (*B. cenocepacia*, *B. cepacia*, *B. multivorans*, *B. dolosa)* were sensitive with a mean MIC of 423 μg/mL. Two of the thirty-five strains of *Proteus* (5.7%) representing two species (*P. vulgaris* and *P. mirabilis*) were sensitive with a mean MIC of 105.7 μg/mL. 

### 3.5. Genera with No Detected Nisin Sensitivity 

The two remaining genera (*Providencia* and *Morganella*) were insensitive to nisin at the highest concentration tested (423 μg/mL). The samples of *Providencia* included three species (*P. rettgeri*, *P. stuartii*, and *P. alcalifaciens*), while *M. morganii* was the sole species representing that genus. 

## 4. Discussion 

The rapidly increasing antimicrobial resistance crisis, coupled with our growing understanding of the critical importance of the human microbiome in supporting human health and the devastating effects antibiotics have on it, and consequently, our ability to fight infections [30], has forced scientists to reconsider their search image in antimicrobial discovery. In place of the conventional, broad-spectrum antibiotics that have served as the cornerstone of bacterial therapeutics, researchers now seek antimicrobials that target specific pathogens while leaving members of the microbiome relatively intact. One appealing family of potential drug candidates that meet this criterion is the bacteriocins, which have repeatedly shown activity against a plethora of multidrug-resistant pathogens, possess both broad and narrow target specificity, and exhibit limited to no toxicity to humans [5,6]. 

One of the more promising bacteriocins in this regard is nisin, which was first discovered in 1920 [31]. Although successfully used as a bio preservative for over fifty years, nisin is now recognized as an attractive drug candidate for the treatment of numerous Gram-positive bacterial infections, including MRSA, MDR *Mycobacterium tuberculosis*, and *Clostridium difficile* [32]. One significant limitation to its potential clinical application is its apparent lack of activity against Gram-negative pathogens [23]. However, a search of the relevant literature revealed very few studies that have deeply screened for nisin activity against Gram-negative genera, and those few that exist included limited numbers of strains per taxon. The present study was designed to provide the first robust screen of nisin sensitivity against some of the more challenging Gram-negative human, animal, and plant pathogens. 

The most compelling result from this study is the surprisingly broad phylogenetic distribution of nisin sensitivity (percent strains sensitive) and activity (mean MIC) detected among the diverse Gram-negative genera tested. Given the relatively few prior reports of nisin activity among Gram-negative bacteria [23] we had assumed that very limited levels of nisin sensitivity would be observed. Not only were the levels significantly higher than anticipated, but the distribution of nisin sensitivity was phylogenetically broad. For example, nisin sensitivity was detected across the breadth of Proteobacteria, including *Escherichia*, *Neisseria* [33] and *Helicobacter*. Coupled with the nisin sensitivity data reported for Gram-positive bacteria [34], it is reasonable to assume that nisin sensitivity is an ancestral state and insensitivity has evolved multiple times in separate lineages. This observation predicts that a diversity of resistance mechanisms may be identified, which has been confirmed for Gram-positive bacteria [34]. With sensitivity being an ancestral trait, a baseline tolerance to nisin is not expected, but rather, unless a resistant mechanism evolves, then the Gram-positive bacteria will be sensitive to nisin making it a valuable treatment option to further investigate.

This study showed that the use of nisin to treat *Helicobacter pylori*, a critically important gastrointestinal pathogen, warrants further study. The six *H. pylori* strains possessed the lowest MIC values of the seventeen genera tested. Two prior studies also report low nisin MICs, ranging from 0.07 to 2.1 μg/mL for a total of six strains [35,36]. *H. pylori* has been identified as a target for worldwide eradication [37] due to its role in gastric cancer. However, current treatments are both costly and only moderately effective, requiring bismuth plus two antibiotics [38]. *H. pylori* is the number one cause of peptic ulcer disease and is the main risk factor for the development of gastric cancer; in fact, to date, it is the only bacteria considered a group one carcinogen by the WHO [39]. Thus, it is worth emphasizing nisin’s efficacy against this critical human pathogen. Nisin has optimal stability at a pH of 3, and thus may display full antimicrobial activity in a harsh stomach environment [40], but that remains to be tested. In this study, we acknowledge that only a limited number of strains were examined, primarily attributed to the well-known challenges associated with maintaining the viability of *H. pylori* in a laboratory setting [41]. However, the findings presented here, when considered alongside the physicochemical properties of nisin and the clinical importance of *H. pylori*, highlight a substantial scope for future investigation into the potential therapeutic applications of nisin. 

The activity of nisin against *A. baumannii* is also noteworthy, as it is on the CDC’s list for high-risk multidrug resistance and its increasing carbapenem resistance. In this study, all 67 strains of *A. baumanii* strains were highly sensitive to nisin. Two prior studies reported that nisin was not active against the two strains tested [42] or had relatively high MICs that ranged between 50 to 100 μg/mL for the 15 strains tested [5], although both reported synergy when nisin was combined with polymyxin B. These studies employed Mueller–Hinton II agar (MHII), rather than the Luria–Bertani agar (LB) employed here. MHII has more cations than LB, against which the positively charged nisin must compete for access to the lipopolysaccharide layer of the outer membrane (Figure 1B). A similar impact of MHII was reported for susceptibility testing of *P. aeruginosa* against gentamicin [43]. Thus, it is possible that the higher nisin MICs detected in prior studies are the result of the growth conditions employed. 

*Erwinia* and *Xanthomonas*, two prominent genera of plant pathogens, showed significant sensitivity to nisin, indicating its potential as a biocontrol agent. *E. amylovora*, the cause of fire blight in fruit trees, and various *Xanthomonas* species, responsible for diseases in diverse crops, rank among the top ten scientifically and economically significant bacterial pathogens in plants [44]. In this study, all 21 strains of *E. amylovora* and all 11 strains of *Xanthomonas* exhibited sensitivity to nisin. These findings highlight nisin’s potential as a tool for managing bacterial plant diseases. Additionally, nisin’s proteinaceous nature allows for biodegradation, distinguishing it from conventional pesticides and antibiotics which persist and accumulate in the environment [45]. Utilizing nisin as a plant biocontrol agent also offers the advantage of reduced harm to pollinators and other wildlife in crop ecosystems, thanks to its favorable toxicity profile [46]. 

*P. syringae* is the most economically important bacterial plant pathogen and could even be considered one of the most successful plant pathogens in agricultural history [44]. Pathovars of *P. syringae* have been identified for nearly all economically important crops, collectively resulting in substantial global economic losses each year [47]. In this study, we tested 14 isolates of *P. syringae* and found an impressive 100% sensitivity to nisin with MICs much lower than that of streptomycin and copper, the premier antimicrobials employed in its biocontrol [48]. A prior study reported an MIC of 75 μg/mL for streptomycin and 375–500 μg/mL for copper against a set of 108 *P. syringae* pv. syringae isolates [49], which is notably higher than the MIC range of 6.6–52.8 μg/mL reported in the present study. Our findings highlight the potential of nisin as a biocontrol agent against *P. syringae*, warranting further exploration. 

*Vibrio* species have emerged as a prominent etiological factor in global seafood-borne illnesses, while directly contributing to substantial economic losses within the aquaculture sector through livestock infection. Human infections caused by *Vibrio* can manifest as anything from the less innocuous gastrointestinal discomfort to severe, life-threatening conditions such as cholera [50]. Likewise, certain species, such as *V. parahaemolyticus*, which is ubiquitous in marine environments, can assume the role of opportunistic pathogens, leading to substantial economic ramifications [51]. A notable illustration of this is the emergence of Acute Hepatopancreatic Necrosis Disease (AHPND) in penaeid shrimp populations in Southeast Asia, which culminated in the partial collapse of shrimp aquaculture industries across multiple countries in 2013 [52]. In the present study, nisin demonstrated remarkable efficacy, with 79.2% of the tested *Vibrio* isolates exhibiting sensitivity. This finding suggests an opportunity for exploring nisin’s potential as a therapeutic agent for human health or as a biocontrol agent in aquaculture. The significance of nisin in aquaculture is further underscored by global initiatives aimed at regulating agricultural antibiotic usage, including the efforts by the WHO and other global entities, the implementation of antimicrobial resistance action plans by major shrimp-producing nations, and the scrutiny surrounding import standards in prominent shrimp-importing countries [53]. While warranted, the industry’s shift away from antibiotic reliance in treating infections like AHPND renders it vulnerable to significant economic losses. Consequently, the outcomes of this study serve as a catalyst for further investigations into harnessing nisin as a sustainable solution to address these challenges. 

Recently, *S. marcescens* has gained attention as an emerging pathogen worldwide, particularly in newborns and patients in intensive care units. Isolates recovered from clinical settings are frequently described as multidrug-resistant [54]. Our screen included 30 isolates of *S. marcescens* and 11 of *S. plymuthica*, of which 58% showed nisin sensitivity. No prior screens of nisin sensitivity against *Serratia* were identified in the literature. The relatively high MICs, ranging from 13.2 to 423 μg/mL, suggest that nisin may not be an appropriate antimicrobial for this genus. However, as is the case with *E. coli*, nisin sensitivity may be enhanced by the addition of ethylenediaminetetraacetic acid (EDTA) [55,56], which is thought to replace divalent cations from their binding sites and reduce the interaction between LPS molecules, thus exposing the lipid II target [57]. Other groups have attempted to bioengineer nisin to obtain higher activity against select Gram-negative pathogens. For example, Li et al. (2018) fused peptide tails to nisin which improved its activity by 4–12-fold against several Gram-negative bacteria, including *Escherichia coli*, *Klebsiella pneumoniae*, *Acinetobacter baumannii*, *Pseudomonas aeruginosa*, and *Enterobacter aerogenes* [17].

*Citrobacter* spp. are emerging as common nosocomial multidrug-resistant pathogens, especially in developing countries. In one study, hospital-acquired urinary tract infections caused by *Citrobacter* spp. account for approximately 9.4% of the total cases [58]. C. *freundii* is now a pathogen of concern as the species has increasingly become MDR [59]. Four species of Citrobacter were included in this study, thirty-three *C. freundii,* thirteen *C. amalonaticus*, two *C. diversus,* and one *C. koseri*, of which 80% were sensitive, with a mean MIC of 267.9 μg/mL. Although these MICs are quite high, it is worth noting that combinations of nisin and conventional antibiotics often result in synergy against Gram-negative genera [60], and as stated above, nisin can be bioengineered to create more effective variants.

Table 1 reports the mean MICs in both μg/mL and nmol/mL units. Mean MICs are reported as the number of strains assayed was too small to yield meaningful MIC_90_ values. The inclusion of MICs in nmol units points out a relevant aspect of nisin activity, bacteriocins are far larger than conventional antibiotics, and thus MIC values of μg/mL provide a slightly skewed metric for comparison. For example, nisin, a 34 amino acid peptide, has a molar mass of 3354.07 g/mol and a mean MIC of 5.6 μg/mL for the panel of *A. baumanii* tested here. Therefore, approximately 1 × 10^15^ molecules of nisin can inhibit the average *A. baumanii* isolate. In contrast, meropenem has a roughly ten-fold lower molar mass (383.5 g/mol) and an MIC_90_ of >8 μg/mL when assessed against a panel of 5478 clinical isolates [61], which translates into 1.26 × 10^16^ molecules, over an order of magnitude more molecules than nisin. While this may seem like a negligible difference, it has significant implications for the potential for toxicity and adverse reactions [62]. 

A large concern with developing antimicrobial treatments is their ability to penetrate and function when used to treat biofilm-forming microbes. Biofilms are complex structures made up of one to several bacterial species that are connected and protected by self-secreted polymeric extracellular substances [63]. Nearly 99.9% of microbes can develop biofilms on biotic and abiotic surfaces making biofilm-penetrating treatments essential in drug development [64]. Known biofilm-producing genera like *Acinetobacter*, *Pseudomonas*, *Klebsiella*, and *Enterobacter* are an increasing threat to public health due to their antibiotic resistance which in part is caused by their biofilm production [65]. While this study did not investigate nisin’s effects against the genera tested in biofilms, but rather in planktonic cells, this is an interest for future studies as nisin’s ability to penetrate a biofilm has been previously seen in Gram-positive genera like *Staphylococcus*, *Streptococcus,* and *Actinomyces* [66], and in synergy with polymyxin B against Gram-negative *Pseudomonas aeruginosa* [67]. Future studies based around biofilm-forming genera, like *Pseudomonas* and *Acinetobacter*, are of interest and the full anti-biofilm capabilities of nisin against Gram-negative pathogens should be investigated. 

This study focused on Gram-negative genera, for which nisin has a far less well-defined mode of action. The outer membrane (OM) of Gram-negative bacteria provides the cell with a permeability barrier against numerous external agents, including antimicrobials. This impermeability has been proposed as the reason for nisin’s poor activity against some Gram-negative genera [32], with the OM preventing access to the inner membrane and nisin’s lipid II target. However, the present study reveals unexpectedly high levels of nisin sensitivity among a phylogenetically broad panel of Gram-negative genera. Gram-negative inhibition occurs via nisin creating a wedge formation in the outer membrane while simultaneously destabilizing the outer membrane by binding to a region of the LPS. Nisin’s LPS binding is competitive with divalent cations, like Mg^+2^ and Ca^+2^, which tighten the OM rather than loosening it and the availability of nisin binding has shown dependency on whether the LPS is rough or smooth [31]. 

The ‘smooth’ or ‘rough’ characterization describes how different LPS structures appear. Smooth LPS has fewer structural irregularities and fewer gaps in its structures. Rough LPS has more ‘texture’, referring to structural irregularities and gaps. The O polysaccharide, also known as the O antigen, is attached to the core oligosaccharide located in the outermost portion of the OM. Smooth LPS has an O antigen attached, while rough LPS lacks the O antigen. Lanne et al. [31] report that the absence of an O antigen in the rough phenotype results in LPS presenting nisin-binding sites located near the membrane surface. They argue that easier accessibility of these sites near the membrane enhances membrane disruption by nisin. Further, they note that deletions in the LPS moieties in *S. typhimurium* resulted in increased nisin sensitivity [63]. Although the smooth versus rough phenotype does explain some of the variation in nisin sensitivity, further investigation is required. For example, two species of *Pseudomonas*, *P. aeruginosa and P. syringae*, were included in this study due to their relevance to plant *(P. syringae)* and human *(P. aeruginosa)* health. They are both known to have relatively high levels of diversity in the O antigen regions of their LPS [68,69], but exhibit significantly different levels of nisin sensitivity, with mean MICS of 27.18 μg/mL (*P. syringae*) and 350 μg/mL (*P. aeruginosa*). Clearly, factors beyond O antigen presentation play a role in determining nisin sensitivity.

Although not a focus of this study, nisin has been examined for synergy with conventional antibiotics. One example is with *P. aeruginosa*, which has a very high mean MIC suggesting nisin may not be an appropriate antimicrobial. However, a synergistic relationship has been observed between nisin and polymyxin B when tested against *P. aeruginosa* strains [70]. Nisin was also shown to act synergistically with polymyxin B when tested against strains of *Acinetobacter baumannii* [5]. Combinations of nisin and ceftriaxone or cefotaxime were shown to be highly synergetic when tested against *Salmonella* strains [71]. A future investigation will focus on those Gram-negative taxa with high mean MICs and explore the synergistic interactions between nisin and a panel of conventional antibiotics. 

A second future avenue of investigation will focus on nisin’s ability to penetrate biofilms. Nisin has been shown to penetrate the biofilms of *P. aeruginosa*, *S. aureus* [13], *A. baumanii* [72], and *S. epidermidis* [73] as well as inhibit biofilm formation in *Listeria* spp. [74] and *Enterococcus faecalis* [75]. A more systematic investigation of nisin’s ability to impact biofilm formation and to inhibit strains in an established biofilm across a large number of Gram-negative taxa is warranted.

## 5. Conclusions

The long history of nisin use in the food industry and the impressive body of knowledge about its structure and function, position nisin as a high-priority option for antibacterial drug development. We have shown here that the use of this peptide can extend beyond Gram-positive genera and includes clinically and economically relevant Gram-negative species as well. This knowledge will also inform efforts at tailoring the structure of nisin and/or providing synergistic supplements (such as EDTA) that increase the availability of lipid II and thus enhance the sensitivity of target pathogens. LPS is posited as a potential “receptor” for nisin, one that can facilitate nisin-induced membrane disruption [31,63]. The importance of these results is enhanced by the very low frequency of resistance to nisin observed in Gram-positive bacteria to date, and the fact that LPS presentation in Gram-negative bacteria is unlikely to be eliminated by simple mutations [31]. This study provides insights into the inhibitory nature of nisin against Gram-negative genera, which opens countless avenues of study through which nisin might be applied. 

## Figures and Tables

**Figure 1 microorganisms-12-01230-f001:**
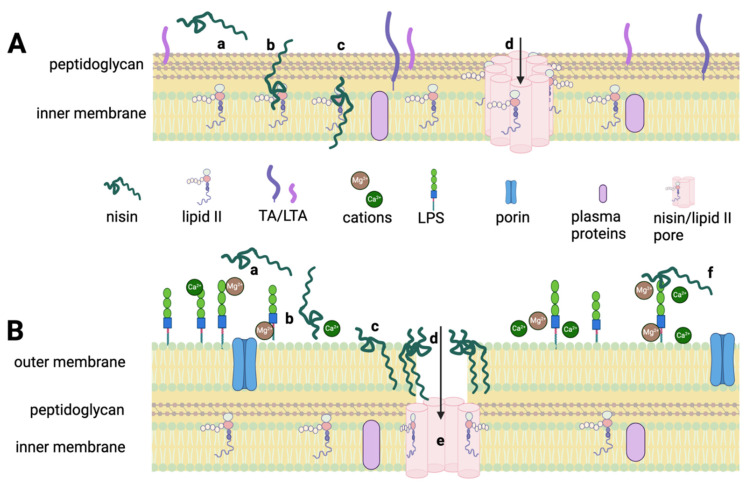
Pathways of nisin inhibition against Gram-positive (**A**) and Gram-negative (**B**) bacteria.

**Figure 2 microorganisms-12-01230-f002:**
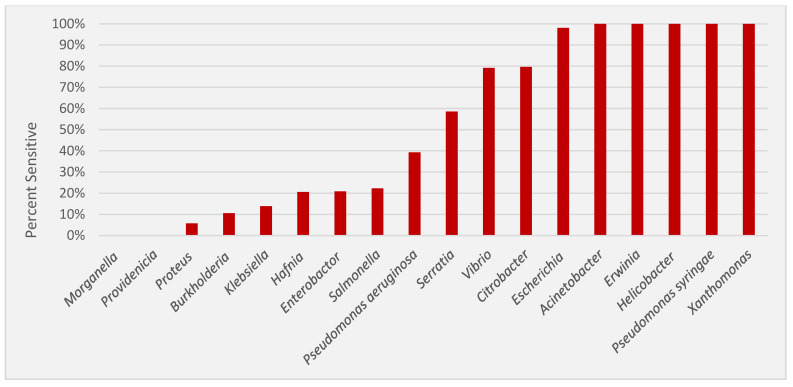
The rank order of nisin sensitivity (from least to highest) for 17 genera of Gram-negative bacteria. Isolates of *P. aeruginosa* and *P. syringae* are provided separately.

**Figure 3 microorganisms-12-01230-f003:**
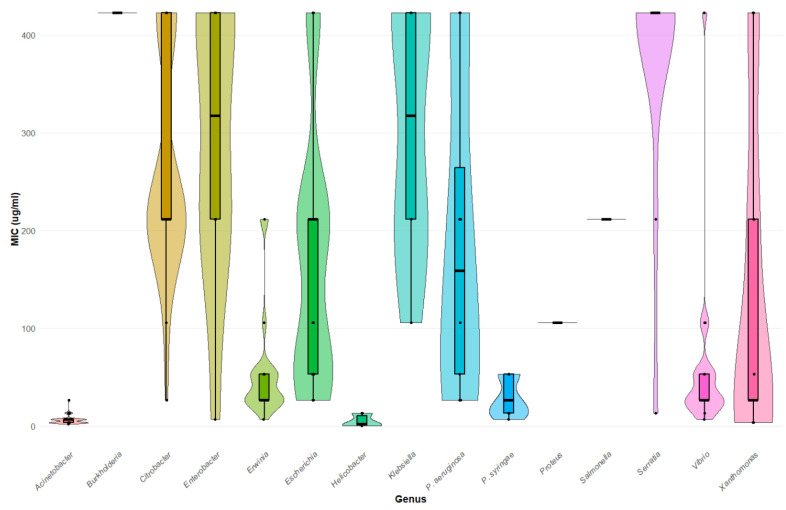
Summary statistics of MIC data for Gram-negative bacteria: Violin plots of MICs for 17 genera of Gram-negative bacteria are provided. MICs are on the y-axis, and genera are indicated on the x-axis. *P. aeruginosa* and *P. syringae* are provided separately. Thin black horizontal lines indicate quartiles; thick black horizontal lines indicate medians; widths of violin plots represent the proportion of isolates for a given MIC, and black dots are where clusters of similar MICs fall.

**Table 1 microorganisms-12-01230-t001:** Nisin activity against 17 genera of Gram-negative bacteria.

Genus	# of Strains	# of Nisin Sensitive (%) *	MIC Range (μg/mL)	MIC Mean (μg/mL)	MIC Mean (nmol/mL)
*Acinetobacter*	67	67 (100%)	1.65–26.40	5.86	1.75
*Burkholderia*	19	2 (10.50%)	423	423	126.12
*Citrobacter*	49	39 (79.60%)	26.40–423	267.85	79.86
*Enterobacter*	48	10 (20.80%)	6.60–423	296.76	88.48
*Erwinia*	21	21 (100%)	6.60–211.50	55.66	16.60
*Escherichia*	51	50 (98%)	26.40–423	171.81	51.22
*Hafnia*	39	8 (20.50%)	105.70–423	347.46	103.59
*Helicobacter*	6	6 (100%)	0.21–13.20	5.12	1.5265
*Klebsiella*	65	9 (13.80%)	105.70–423	304.03	90.65
*Morganella*	13	0 (0.00%)	NS	NS	NS
*Proteus*	35	2 (5.70%)	105.70	105.70	31.51
*Providencia*	13	0 (0.00%)	NS	NS	NS
*Pseudomonas*	44	38 (86.40%)	6.60–423	92.50	27.58
*Salmonella*	9	3 (33.33%)	211.50	211.50	63.06
*Serratia*	41	24 (58.50%)	13.20–423	371.23	110.68
*Vibrio*	24	19 (79.20%)	3.30–423	60.13	17.93
*Xanthomonas*	11	11 (100%)	13.10–423	130.36	38.87

* Sensitivity is based on the tested concentrations that had a maximum of 423 μg/mL or 126.12 nmol/mL. NS = not sensitive to the range of concentrations tested.

## Data Availability

The data underlying this article will be shared on reasonable request to the corresponding author.

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
