# Peer review of "Nisin Inhibition of Gram-Negative Bacteria"

_microorganisms, 2024, doi:10.3390/microorganisms12061230_

Round 1

Reviewer 1 Report

Comments and Suggestions for Authors

Author Response

We wish to thank the reviewer for taking the time to point out several ways in which our manuscript might be improved. We have responded to each of the suggestions below.

  1. The manuscript has been submitted to the special issue “The Potential of Antimicrobial Activity and Antibiofilm Activity of Bacteriocins” but is contains no biofilm experimentation. Biofilms are a major concern for food safety and agricultural infections. The potential of Nisin to prevent biofilm formation should be assessed for select bacteria genera. 4 genera; 2 sensitive and 2 non-sensitive should be assessed. Nisin maybe able to reduce biofilm formation in species that is insensitive to it, thereby still indicating its potential use if used in combinational therapy. We agree that the ability of nisin to impact biofilm formation is relevant (whether a species is sensitive or insensitive to the drug in planktonic state), however it is beyond the scope of this manuscript. We have included a new paragraph in the discussion session that reports the current information about nisin's ability to impact biofilms, which will set the stage for our next study.
  2. The Fractional Inhibition Concentration (FIC) should be determined for Nisin with a select antibiotic against 2 insensitive genera to illustrate if Nisin can work additively or synergistically with other antimicrobials against bacteria that is insensitive to it. We agree that the ability of nisin to work synergistically with other drugs  is relevant, however we intent to conduct a similarly extensive study of this factor as a completely separate study. We feel it is beyond the scope of this manuscript to include a small sample of that sort of data, since our goal is to provide large sample sizes and a robust assessment of synergy with numerous types of antibiotics. We have included several sentences about the potential for synergy to set the stage for this future study.
  3. In the data quantification section, it is indicated that MIC ≤ 26.4 μg/ml indicated high nisin activity and MIC ≥ 211.5 μg/ml corresponded with low activity. These values seem arbitrary, was there a standard or reference guide used to break up the range of MICs into these different categories? The reviewer is correct that these values are arbitrary. Unfortunately, there is no standard we could apply to nisin concentrations and we examined the overall pattern of sensitivities and chose these values as most appropriate. We have included a sentence to explain that these values are arbitrary so that it will be clear to readers.

Reviewer 2 Report

Comments and Suggestions for Authors

It is excellent work, innovative and very useful.

I have a few questions:

1. Why did you add such a large amount of overnight culture to the wells of the microtiter plates?

2. Why did you present results for genera of bacteria in Figure 1 and 2, but singled out only two species of Pseudomonas.

3. The conclusion as a whole is missing at the end. Perhaps the last paragraph could stand out that way.

Author Response

We wish to thank this reviewer for their review, we appreciate their comment on the innovation and utility of our data. They propose several questions we will respond to below.

1. Why did you add such a large amount of overnight culture to the wells of the microtiter plates? The volume of overnight culture was chosen to ensure that we added the precise concentration of cells required by the CLSI standards.

2. Why did you present results for genera of bacteria in Figure 1 and 2, but singled out only two species of Pseudomonas. The two genera of Pseudomonas, P. aeruginosa and P. syringae, are involved in both human (P. aeruginosa) and plant (P. syringae) disease. We felt it was important to separate those two genera so that they could be discussed separately in the conclusions.

3. The conclusion as a whole is missing at the end. Perhaps the last paragraph could stand out that way. We have revised the manuscript to include a concluding paragraph as the reviewer suggests.

Round 2

Reviewer 1 Report

Comments and Suggestions for Authors

The authors investigated the potential of use of Nisin across a series of more than 15 gram-negative bacteria genera. The purpose of the study was to demonstrate that Nisin which has traditionally been thought to be only effective against gram-positive bacteria can be effective against certain gram-negative genera such as Acinetobacter, Helicobacter, Pseudomonas and Xanthomonas. The authors are encouraged to check over the manuscript for grammatical, spelling and citation errors.